# OpenReview forum: "Efficient Sequence Packing without Cross-contamination: Accelerating Large Language Models without Impacting Performance"
_ICLR.cc/2023/Conference — Submitted to ICLR 2023_

### Official Review · Reviewer_1zai · 2022-10-24

**Confidence:** 3
**Correctness:** 3
**Technical Novelty And Significance:** 2
**Empirical Novelty And Significance:** 2
**Recommendation:** 6

**Clarity, Quality, Novelty And Reproducibility:**

Questions:
- Why use BERT as a baseline experimental setup compared to other alternatives such as T5 or roberta?
- Could you explain or justify what you mean:
    - Page 1: “Its effect is enhanced by the offline dataset generation process which, in BERT, attempts to pack together sentences so as to fill the sequence length as completely as possible”.
    - Page 5: “However, this approach might not be desirable as it might imply under-utilizing the memory/compute, especially if the micro batch size needs to be reduced.” -> I did not understand the under-utilization argument.
    - Page 8: “Packing slight violates the i.i.d. Assumption of the data”.
- Terms I had difficulty in understanding:
    - Figure 1: how is the theoretical speedup computed?
    - Figure 3: What is `beta`?
    - Figure 4: what is the training accuracy?
- Presentation suggestion: While wall time is a very useful metric to track, a lot of the engineering challenges when training very large llms could be summarized as “optimizing flops/second throughput”. Presenting that along with iteration count and time would give a more accurate picture of the problem.


**Strength And Weaknesses:**

Strengths:
- The problem is well-posed and relatively well-motivated. This is an important problem that has consequences on a large class of current training schemes and models in NLP but also consequences in training practices outside of NLP.
- Experiments are relatively straightforward and well constructed, and results are convincing.

Weaknesses:
- Usage of BERT as a baseline is not well motivated (does it have to do with MLPerf benchmark?). More specifically, several choices in BERT training have been questioned in the RoBERTa reproduction such as the need for 2 phase training (first on short sequences and then longer sequences), the very big batches, the NSP auxiliary loss… I think the experimental setup could be considerably simplified (and more actual).
- Downstream evaluation performance is somehow weak and not exhaustive enough (only squad performance is reported). I think a minimum of 3 qualitatively different tasks would be enough to convince the reader that the trainings are equivalent. Matching training curves (Figure 3 and 4) seems like only a “sanity check”.


**Summary Of The Paper:**

The authors are interested in studying the impact of the packing strategy on BERT’s training and performance.

The authors first observe that the variation in sequence lengths in nlp datasets used for pre-training (such as Wikipedia) can yield a very large amount of padding tokens (50% reported) which means a lot of compute is wasted on padding tokens.

They propose two heuristics for creating packed sequences taking inspiration from the operational research literature which can scale to very large datasets.

By adjusting the positional embeddings, the attention masks, and losses, the authors show that the packed sequence training is equivalent to the original non-packed BERT training both in terms of downstream performance and training convergence.


**Summary Of The Review:**

Overall, this a solid study of well-posed problem, with limited technical and empirical novelty,

---

> ### Author Response · Authors · 2022-11-17
> **Response to Reviewer 1zai**
>
> We thank the reviewer for carefully reading our manuscript and for bringing up several insightful comments which we feel have helped us improve its quality. We were also encouraged by the Reviewer’s comments: “The problem is well-posed and relatively well-motivated”, “experiments are relatively straightforward and well constructed” and “results are convincing”.
>
>
>
> Let us clarify the weaknesses addressed by the reviewer:
>
> - It’s true that our approach is focused on BERT: this is because BERT, contrary to other models like RoBERTa, GPT-3 or PaLM, can benefit from removing the padded tokens that represent up to 50% of all Wikipedia tokens. The other models allow for larger sequence lengths and concatenation of sequences, thus removing the padding present in the dataset without need of packing. However, BERT is still the most popular and deployed NLP model, and we believe that our packing solution has clear benefits for the practitioners deploying it. Larger models attract a lot of attention and push the state of the art in NLP, but are not that widespread and deployed due to their greater computational cost and memory size. Also, the other models can also benefit from our algorithm modifications to avoid contamination between packed sequences.
>
> - We acknowledge that more than one fine-tuning validation tasks would be more solid, and will incorporate them in the next version of the manuscript. Concerning the training plots in figures 3 and 4, they play an important role in comparing the model and hyperparameter changes detailed in subsections 3.2 and 3.3. Without such changes, the training plots clearly show degradation.
>
> Let us now clarify the remaining questions:
>
> - Page 1: “Its effect is enhanced by the offline dataset generation process which, in BERT, attempts to pack together sentences so as to fill the sequence length as completely as possible” -> This refers to the way the BERT Wikipedia dataset is generated (see reference [8] for details). Basically, the original paper in reference [7] mentions the word “pack” for their approach of combining two segments that might or might not be consecutive for the NSP. This mechanism might be also motivated by the goal to try to fill up the max sequence length as much as possible. Despite it, we show that padding represents 50% of all tokens for sequence length 512. We use this part to counter any argument that BERT is already doing packing.
>
> - Page 5: “However, this approach might not be desirable as it might imply under-utilizing the memory/compute, especially if the micro batch size needs to be reduced.” -> To fully utilize a particular accelerator, one needs to optimize the batch size to fit within the memory of the accelerator. The larger the batch size you can fit, the faster the model will train. What we mean by under-utilising is that, if a particular model has already been optimized to run at a certain batch size in a specific accelerator, reducing that batch size means that there will be memory left that is under-utilized. We can see the confusion here and will rephrase it.
>
> - Page 8: “Packing slight violates the i.i.d. Assumption of the data” -> This is because we are altering the i.i.d. order of sequences that the original dataset satisfies, by concatenating certain sequences together. However, our validation experiments prove that convergence is not degraded.
>
> - Figure 1: how is the theoretical speedup computed? -> Our theoretical speedup computation comes from the fact that, since padding represents 50% of all tokens, after removing it we need half the sequences to contain all the real tokens, and this leads to a theoretical speedup of two. So it is computed simply from the padding percentage in the original dataset, reported in section 2.
>
> - Figure 3: What is beta? -> It is the decay parameter of the LAMB optimizer. The reviewer can find more details about it in Appendix D.
>
> - Figure 4: what is the training accuracy? -> it is the average of the MLM and NSP accuracies, computed as in the original paper of BERT.
>
> - Thanks for the presentation suggestion, we can see how providing flops/second throughput would add value to the manuscript. We will aim to add it in the next version of it.

---

### Official Review · Reviewer_PH3R · 2022-10-24

**Confidence:** 3
**Clarity, Quality, Novelty And Reproducibility:** See strengths and weaknesses.
**Correctness:** 3
**Technical Novelty And Significance:** 2
**Empirical Novelty And Significance:** 2
**Recommendation:** 5

**Strength And Weaknesses:**

Strengths:

1. The proposed techniques are straightforward to implement, and they work well.

2. The experiments are solid, with particular attention to details such as the effect of breaking the iid assumption when packing.

Weaknesses:

1. The contribution seems a bit thin. Although two algorithms are proposed, the simpler of the two (sort sequences by length, allocate by decreasing length to batches with the most space remaining) basically works as well as the more complex matrix formulation, and doesn’t come with the additional restriction of max 3 sequences / batch due to computational constraints. That algorithm is quite obvious, as are the techniques to modify position embeddings and attention to ensure sequence independence.

2. The work is mostly relevant to models such as BERT, which use sentence inputs. More recent models such as GPT-3 and PaLM use much longer sequences, and avoid padding by just concatenating and truncating documents. Larger capacity models also mean that special masks aren’t necessary to distinguish examples, as they can learn to split on just a separator token.

3. Although the paper was quite clear for the most part, it would benefit from a distinct problem definition at the beginning of section 3 that defines terms and gives the inputs and outputs from the algorithm. Some of the descriptions were a bit unclear, e.g. in 4.1 it would be good to have a better definition of sorted batching in the main paper, and also an explanation of how greedy is different from packed BERT.

**Summary Of The Paper:**

This paper proposes two algorithms for packing multiple token sequences into batches so as to minimize the amount of padding required. It also describes Transformer-specific techniques for ensuring that sequences within a packed batch are handled completely independently (no information leaks across batches). Experiments with BERT show speedups of almost 2x compared to unpacked batches, with no degradation in model performance.


**Summary Of The Review:**

I am leaning slightly against acceptance because this seems to be mostly an engineering contribution, and one that is more relevant to older, smaller models. Although such models are still in common use, the training costs of more recent large LMs are so much bigger that they make gains on smaller models seem less important.

---

> ### Author Response · Authors · 2022-11-17
> **Response to Reviewer PH3R**
>
> We thank the reviewer for carefully reading our manuscript and for bringing up several insightful comments which we feel have helped us improve its quality. We were also encouraged by the Reviewer’s comments: “The proposed techniques are straightforward to implement, and they work well” and “The experiments are solid”.
>
> Let us clarify the weaknesses addressed by the reviewer:
>
> - Concerning the novelty, we believe that the impact of the contribution is reflected in recent adoption by NVIDIA, Graphcore and Intel of our approach as the standard for performance benchmarking in NLP.  Essentially, our claim is that this approach is obvious now primarily because of the time delay between its implementation and the submission of this paper, but that nevertheless it is valuable to the community to publish in archival format the analysis of the now-standard technique. Moreover, other related techniques are mainly suited towards GPUs, whereas our algorithm is accelerator-agnostic. More details about related work are provided in Appendix C.
>
> - The comparison of the two packing algorithms is displayed in table 1. The reviewer is right in that the simpler method (denoted as SPFHP) can reach an efficiency of 99.6% for a really large number of sequences per pack, and that efficiency is comparable to the more complex method (denoted as NNLSHP), which has the restriction of a maximum of 3 sequences per pack. However, we pick NNLSHP because even with only 3 sequences per pack it is 1) slightly more efficient reaching 99.7% and 2) has less computational overhead (4.3% vs 4.5% of SPFHP with max depth). In any case, we acknowledge that some practitioners will prefer the simpler method for convenience and will clarify it in the manuscript.
>
> - It’s true that our approach is focused on BERT: this is because BERT, contrary to other models like RoBERTa, GPT-3 or PaLM, can benefit from removing the padded tokens that represent up to 50% of all Wikipedia tokens. The other models allow for larger sequence lengths and concatenation of sequences, thus removing the padding present in the dataset without need of packing. However, BERT is still the most popular and deployed NLP model, and we believe that our packing solution has clear benefits for the practitioners deploying it. Larger models attract a lot of attention and push the state of the art in NLP, but are not that widespread and deployed due to their greater computational cost and memory size.
>
> - Whereas BERT obtains a major speed-up from the removal of padding tokens with our approach, other models will get increased predictive accuracy by avoiding contamination between sequences. It would be great, if the reviewer could provide a reference for the claim “they can learn to split on just a separator token”. If this were true, the performance without mixing documents and with concatenating them with a separator token should be the same. As highlighted in our paper, this is clearly not the case for the RoBERTa algorithm. The main reason why RoBERTa performs better than BERT at the end is that it does not have the padding tokens and thus trains on more than 2x tokens, which is not a fair comparison.
>
> - Good point about clarifying the problem definition in sections 3 and 4.1 — we see how this can cause confusion and will rephrase it and add further detail.

---

> > ### Comment · Reviewer_PH3R · 2022-11-22
> > **Response read**
> >
> > Thanks for the response. I don't think we have any major points of disagreement, but I'm still not able to strongly endorse this paper for ICLR, mostly because of the nature of the contribution.
> >
> > Example of my assertion about the separator token: Chowdhery, Aakanksha, et al. "Palm: Scaling language modeling with pathways." arXiv preprint arXiv:2204.02311 (2022), pg 10, "Sequence length".  They don't explicitly state that no attention mask is used, but this is the implication. This is a minor point, however, and not central to my review.

---

### Official Review · Reviewer_tpoA · 2022-10-25

**Confidence:** 3
**Clarity, Quality, Novelty And Reproducibility:** Please refer to Detailed Comments.
**Correctness:** 3
**Technical Novelty And Significance:** 2
**Empirical Novelty And Significance:** 2
**Recommendation:** 3

**Strength And Weaknesses:**

Strengths:
- This paper designs shortest-pack-first histogram-packing (SPFHP) and non-negative least-squares histogram-packing (NNLSHP) algorithms to avoid the padding tokens and speed up the model training.

Weaknesses:
- The technical contribution of this paper is limited. Packing is a common technique and has been widely adopted in the official deep learning frameworks, including tensor2tensor and Fairseq. This seems to be a system paper and the contribution might be not significant enough to publish at ICLR.
- A simple packing technique is to sort sequences by length and merge near-length sequences into one mini-batch. Then it further leverages dynamic batch size to maintain a similar token number for each mini-batch, which significantly speeds up the model training. This paper lacks detailed comparisons with this simple baseline.
- Currently, dynamic sequence length is supported in CUDA libraries, which may limit the application of the proposed algorithms.

Detailed Comments:

Overall, this paper is easy to follow and this idea is straightforward. But the technical contribution of this paper is limited. This work seems to be a system paper and the contribution might be not significant enough to publish at ICLR.

Packing is a common technique and has been widely adopted in the official deep learning frameworks, including tensor2tensor and Fairseq. For example, a widely used packing technique is to merge near-length sequences into one mini-batch and then adopt dynamic batch size to maintain a similar token number for each mini-batch. It could achieve significant speedup during model training and there is no detailed comparison with this baseline. In addition, the dynamic sequence length of RNN or Transformer is supported in CUDA libraries, in which the CUDA libraries avoid the computation of padding tokens when given sequence length. This way limits the application of the proposed algorithms and this paper lacks correspondent comparisons between the proposed method and dynamic sequence length.


Questions for the Author(s):
- I would like to know the details of the sorted method in the paper. Does it adopt dynamic batch size to maintain a similar token number for each mini-batch?
- Currently, dynamic sequence length is supported in CUDA libraries. How about the differences between the proposed method and dynamic sequence length?


**Summary Of The Paper:**

This paper explores efficient packing methods for training sequences of BERT,  which avoids the padding tokens to speed up the training.
The authors introduce shortest-pack-first histogram-packing (SPFHP) and non-negative least-squares histogram-packing (NNLSHP) algorithms, which are shown to be straightforward to implement and have little impact on model performance.  Experiments on the Wikipedia dataset show that the proposed method can achieve 2x speedup while achieving similar training loss.



**Summary Of The Review:**

This paper designs shortest-pack-first histogram-packing (SPFHP) and non-negative least-squares histogram-packing (NNLSHP) algorithms to avoid the padding tokens and speed up the model training. But the technical contribution of this paper is limited. This work seems to be a system paper and the contribution might be not significant enough to publish at ICLR. Some strong baselines should be considered.

---

> ### Author Response · Authors · 2022-11-17
> **Response to Reviewer tpoA**
>
> We thank the reviewer for carefully reading our manuscript and for bringing up several insightful comments which we feel have helped us improve its quality. Let us clarify the weaknesses addressed by the reviewer:
>
> - The reviewer is correct that there exist alternative packing techniques in the literature (faster transformer, fairseq or tensor2tensor to name but a few), and several of these are compared in our work (see figure 5 or Appendix C).  We believe that the impact of the contribution is reflected in recent adoption by NVIDIA, Graphcore and Intel of our approach as the standard for performance benchmarking in NLP.  Essentially, our claim is that this approach is obvious now primarily because of the time delay between its implementation and the submission of this paper, but that nevertheless it is valuable to the community to publish in archival format the analysis of the now-standard technique.
>
> - The simple technique the reviewer points out is already included in our paper: in subsection 4.4 we compare it to our packing algorithm. Such technique is only available in GPUs and wouldn’t be efficient for other accelerators. It also presents load-balancing issues that our approach overcomes. We refer the reader to Figure 5 for a graphical comparison of such technique present in Effective Transformer and our packing approach.
>
> - Our packing approach, contrary to CUDA approaches, is accelerator-agnostic and can be reproduced with the code attached in Appendix U. To add more, even NVIDIA has started using our packing approach in their MLPerf submission. This indicates that our packing algorithm presents advantages that are, at least, complementary to the dynamic sequence length the reviewer points out.
>
> - Concerning the comparison with dynamic batch size, our packing algorithms remove the padding present in the dataset to a great extent (see table 1). For our preferred method of NNLSHP with pack depth 3, padding is reduced by 99.7%. This means that no dynamic batch size is used since there are almost no padding tokens and all tokens are real. Consequently, each batch and mini-batch has almost the same number of real tokens.
>
> - For the sorted batching approach, we used a standard batch size of 3000 without load balancing of tokens. We only grouped sequences of the same size together. A similar approach has been used in the past at MLPerf, but participants refrain from it because it impacts convergence speed. It also requires significant optimization efforts in low level code to obtain speed-ups from it. For those two reasons, it is no longer considered practical.

---

> > ### Comment · Reviewer_tpoA · 2022-11-22
> > **Response**
> >
> > Thanks for your response and clarification. I keep my score unchanged.

---

### Official Review · Reviewer_cSJD · 2022-10-25

**Confidence:** 4
**Correctness:** 2
**Technical Novelty And Significance:** 2
**Empirical Novelty And Significance:** 2
**Recommendation:** 3

**Clarity, Quality, Novelty And Reproducibility:**

- The paper is clearly written.
- The novelty and execution are below average for top ML venues.
- The authors provide enough experimental details. But since the experiments are done on special hardware that most researchers don't have access to, I'm concerned about its reproducibility.

**Strength And Weaknesses:**

Strength:
- The paper is fairly clear and well-motivated.
- The proposed technique can be useful for most practitioners.
- General guidelines for adjusting existing hyperparamters.

Weaknesses:
- As a machine learning research paper, the technical contrition is thin. Some venus on MLM systems might be a better fit.
- I’m not an expert on this, but I feel that existing work must have tried to address this issue, but the paper does not compare to any baseline.
- The claim is not well supported by the evidence: (1) the paper claims “without impacting performance,” but the only evaluation on accuracy seems to be MLM training accuracy (in terms of word prediction?); (2) the paper fails to acknowledge that the proposed method may not work for neural models other than transformers (e.g., I cannot figure out how to make this work for RNNs)
- The results are based on IPU, a kind of accelerator that most practitioners probably don’t have access to. This casts some shadow on the paper’s reproducibility, and limits its impact.
- It is not entirely clear how the “theoretical speedup” is determined. Is it simply 1 divided by the ratio of the real tokens? I don’t think this is correct: transformers have quadratic-complexity components. As an illustrating example, I do not think the time of processing a 512-length sequence is twice that of a 256-length one.

Details and further comments:
- End of section 2: please 2.2$\times$, instead of 2.2$x$.
- 4.2.1: what is the task? Is it word prediction on the training data (that no one cares about)? I would strongly encourage evaluate on downstream tasks.
 - I would appreciate it if the authors can try (1) other tasks such as text generation, and (2) other pretrained LMs.


**Summary Of The Paper:**

This paper tackles batching and padding, a core procedure in training machine learning models. It argues that conventional solutions (padding to the max sequence length in a minibatch) may result in 50% of the training data being paddings, substantially wasting the compute. It proposes several techniques to address this problem: better packing of the training sequences, modifications to the transformer architectures, and adjustments to the established hyperparameters. Experiments of training the BERT model show that substantial speedup can be achieved with the proposed technique.


**Summary Of The Review:**


Summary:
- While the proposed method can be useful, further evaluation is needed to validate its effectiveness. I encourage the authors to consider submitting to some ML system venues, which can be a better fit than ICLR.

---

> ### Author Response · Authors · 2022-11-17
> **Response to Reviewer cSJD**
>
> We thank the reviewer for carefully reading our manuscript and for bringing up several insightful comments which we feel have helped us improve its quality. We were also encouraged by the Reviewer’s comments: “The paper is fairly clear and well-motivated” and “The proposed technique can be useful for most practitioners”.
>
> Let us clarify the weaknesses addressed by the reviewer:
>
> - The reviewer is correct that there exist alternative packing techniques in the literature (faster transformer, fairseq or tensor2tensor to name but a few), and several of these are compared in our work (see figure 5 or Appendix C).  We believe that the impact of the contribution is reflected in recent adoption by NVIDIA, Graphcore and Intel of our approach as the standard for performance benchmarking in NLP.   Essentially, our claim is that this approach is obvious now primarily because of the time delay between its implementation and the submission of this paper, but that nevertheless it is valuable to the community to publish in archival format the analysis of the now-standard technique.
> - The validation of our packing algorithm is indeed focused on BERT, whose loss is a combination of MLM and NSP task (see figure 3). This is the classical training procedure for BERT. In addition, we also fine-tune BERT on the SQuAD dataset and show that there’s no impact in prediction (see table 3). We reckon that these validations are enough to prove that packing doesn’t degrade the performance of BERT.
>
> - We haven’t yet thought the application of packing to RNNs fully through but it would probably require some unchanged reset token that resets the hidden states to their original values. This probably also means a modification of the backpropagation through time.
>
> - In the introduction, we specify that our packing approach only applies to transformer models: “in this paper, we formally frame the packing problem in transformer-based models...”. The reviewer is right in that packing may not work for other models other than transformers. We’ll add a note to clarify this.
>
> - The results are obtained in an IPU accelerator, but reproducibility is not at risk because 1) our algorithm is accelerator-agnostic as explained in the introduction of subsection 4.1 and 2) all the code is available in Appendix U and can be reproduced with any accelerator, and 3) has been already reproduced by NVIDIA, Intel and Graphcore.
>
> - The time to process a sequence of 512 tokens is the same when the sequence has half real tokens and half padding, or when it has all real tokens. With our packing approach, we concatenate various sequences within those 512 tokens, removing padding as much as possible. Our theoretical speedup computation comes from the fact that, since padding represents 50% of all tokens, after removing it we need half the sequences to contain all the real tokens, and this leads to a theoretical speedup of two. Note that the quadratic computational cost with respect to sequence length is important when comparing different sequence lengths, but when we pack we don’t vary the sequence length — we just remove the padding and add more real tokens. Depending on the chosen graph optimization and computation strategy (e.g., no XLA backend optimization), different sequence lengths can be processed to take advantage of shorter sequence length as addressed independently in Section 4.4. However, this is a hardware specific solution that is not always advantageous as shown NVIDIA’s usage of our technique.
>
> - Thanks for the further comments. We see how they can cause confusion and will rephrase them in the manuscript.

---

> > ### Comment · Reviewer_cSJD · 2022-11-18
> > **Response**
> >
> > Thanks for the clarification. I keep my score unchanged.

---

### Author Response · Authors · 2022-11-17
**General response to initial reviews**

These reviews make great suggestions for clarification and improvement of this paper.  We believe we have answered the various technical questions and justified the contribution of our paper. We are encouraged that the reviewers generally agree that packing, while looking like an engineering detail, in fact touches interesting research questions. As mentioned in some of our answers, this is a methodology that is exemplified for the most part in codebases, with little analysis, ablation, or theoretical underpinnings.  We believe that aggregating this diffused knowledge in a coherent framework is a benefit to the community, and this is on top of the novelty of our specific approach, which is already acknowledged by its adoption by other teams in the community. For example, NVIDIA, Graphcore and Intel have recently adopted our approach as the standard for performance benchmarking in NLP.

---

### Decision · Program_Chairs · 2023-01-20

**Decision:**

Reject

**Justification For Why Not Higher Score:**

This seems to be a system paper and the contribution might be not significant enough to publish at ICLR.

**Justification For Why Not Lower Score:**

NA

**Metareview: Summary, Strengths And Weaknesses:**

Summary:

This paper proposes methods to reduce the amount of padding while packing multiple sequences in a batch while training BERT.  Multiple technologies are developed to tackle this problem.  Experiments show 2x speedup obtained by using this techniques without degradation in model performance.

Strengths:

The paper addresses a techinical problem which is practically important in the training of pretrained language models.  The paper is well motivated.  The writting is clear and the methods are simple and easy to follow.  The results are convincing.

Weaknesses:

This seems to be a system paper and the contribution might be not significant enough to publish at ICLR.
Lack of comparisons to some simple ideas.
Other language models like GPT are not discussed.


**Summary Of Ac-Reviewer Meeting:**

NA